# Identification of Prognostic and Chemopredictive microRNAs for Non-Small-Cell Lung Cancer by Integrating SEER-Medicare Data

**DOI:** 10.3390/ijms22147658

**Published:** 2021-07-17

**Authors:** Qing Ye, Joseph Putila, Rebecca Raese, Chunlin Dong, Yong Qian, Afshin Dowlati, Nancy Lan Guo

**Affiliations:** 1Mary Babb Randolph Cancer Center, West Virginia University Cancer Institute, West Virginia University, Morgantown, WV 26506, USA; qiye@mix.wvu.edu (Q.Y.); jpputila@gmail.com (J.P.); rebecca.raese@gmail.com (R.R.); lindadong2004@yahoo.com (C.D.); 2National Institute of Occupational Safety and Health, 1095 Willowdale Road, Morgantown, WV 26505, USA; yaq2@cdc.gov; 3Case Comprehensive Cancer Center, Case Western Reserve University, 10900 Euclid Ave., Cleveland, OH 44106, USA; axd44@case.edu

**Keywords:** microRNA (miRNA), prognosis, chemoresponse, non-small-cell lung cancer (NSCLC), CRISPR-Cas9, RNA interference (RNAi)

## Abstract

This study developed a novel methodology to correlate genome-scale microRNA (miRNA) expression profiles in a lung squamous cell carcinoma (LUSC) cohort (*n* = 57) with Surveillance, Epidemiology, and End Results (SEER)-Medicare LUSC patients (*n* = 33,897) as a function of composite tumor progression indicators of T, N, and M cancer stage and tumor grade. The selected prognostic and chemopredictive miRNAs were extensively validated with miRNA expression profiles of non-small-cell lung cancer (NSCLC) patient samples collected from US hospitals (*n* = 156) and public consortia including NCI-60, The Cancer Genome Atlas (TCGA; *n* = 1016), and Cancer Cell Line Encyclopedia (CCLE; *n* = 117). Hsa-miR-142-3p was associated with good prognosis and chemosensitivity in all the studied datasets. Hsa-miRNA-142-3p target genes (*NUP205*, *RAN*, *CSE1L*, *SNRPD1*, *RPS11*, *SF3B1*, *COPA*, *ARCN1*, and *SNRNP200*) had a significant impact on proliferation in 100% of the tested NSCLC cell lines in CRISPR-Cas9 (*n* = 78) and RNA interference (RNAi) screening (*n* = 92). Hsa-miR-142-3p-mediated pathways and functional networks in NSCLC short-term survivors were elucidated. Overall, the approach integrating SEER-Medicare data with comprehensive external validation can identify miRNAs with consistent expression patterns in tumor progression, with potential implications for prognosis and prediction of chemoresponse in large NSCLC patient populations.

## 1. Introduction

Lung cancer remains the leading cause of cancer-related death in the US with a 5 year survival rate of 21.7% [1], due in part to the minimal response to chemotherapy and metastasis [2,3,4]. According to the National Cancer Institute (NCI) Surveillance, Epidemiology, and End Results (SEER) [1], 56% of lung cancer cases are at the distant stage, meaning that cancer has metastasized at the time of diagnosis. Non-small-cell lung cancer (NSCLC) accounts for 84% of lung cancer cases. Major histological subtypes of NSCLC include lung adenocarcinoma (LUAD), squamous cell carcinoma, and large-cell carcinoma. According to the current practice guidelines, NSCLC patients with stage II and above receive chemotherapy, with additional radiation for stage III and IV patients [5]. While adjuvant chemotherapy of stage II and stage III disease has resulted in 10–15% increased overall survival [6], the prognosis for resectable NSCLC remains poor [7]. Patients with the same cancer stage and tumor grade may have markedly different responses to a given treatment, indicating that these morphological tumor markers alone are not sufficient for the selection of an appropriate course of therapy. The use of molecular biomarkers to enhance treatment selection is a promising avenue for improving patient outcomes.

miRNAs are small noncoding molecules (~22 nucleotides) that function in RNA silencing and post-transcriptional regulation of gene expression [8]. MiRNAs are promising biomarkers for both prognosis and prediction of therapeutic response in multiple cancer types [9,10,11]. There are several advantages of using miRNAs in diagnostics/prognostics, such as their presence in circulating plasma [12,13] and the greater stability in prepared tissue samples relative to mRNA [14,15], including formalin fixation [16]. The use of miRNA biomarkers in the selection of a biologically appropriate treatment has the potential to improve clinical outcomes by determining the best application of therapeutic regimens, as well as the elucidation of post-transcriptional regulatory pathways and networks which may aid in the development of novel intervention strategies [17,18,19]. Numerous studies have shown that miRNA biomarkers could potentially be used in the diagnosis and prognosis of lung cancer [9,11,20,21,22,23,24]. To date, there are no clinically applicable miRNAs for NSCLC prognosis and treatment selection.

With the increasing use of genome-scale profiling technologies in clinical and translational research, it is still infeasible to perform transcriptional profiling in hundreds of thousands of patients of a cancer type for the development of novel biomarkers due to the required costs, time, and infrastructure. Current biomarker studies identify/validate candidate genes from clinical cohorts of a limited number of patient samples, and then leverage public data in consortia such as The Cancer Genome Atlas (TCGA) to confirm the candidate genes. There are several issues with these common practices. Firstly, published individual patient cohorts may not have complete treatment information, and the number of patients in specific treatment categories is very small, making it difficult to evaluate predictive biomarkers of therapeutic response. Secondly, some sequencing facilities may not have access to patient outcomes on the samples they have sequenced to conduct genome-wide association studies (GWAS). Thirdly, on the other hand, large-scale patient electronic medical records (EMRs) of hospitals or cancer registries have sufficient number of patients with comprehensive clinical information, but do not have patient genome-scale profiles in general for GWAS. Novel methodologies are needed to fill the gap between genomic/transcriptomic profiles in singular patient cohorts and large-scale EMRs to estimate the applicability of genomic/transcriptomic biomarkers in general patient populations.

The SEER database is an aggregate of registry data from specific geographic areas covering approximately 26% of the US population [25]. The linked SEER-Medicare data are well annotated and ready for computational analysis without natural language processing. There has been reported success in identifying chemoresponse predictive genes by correlating mRNA expression profiles in solid tumors of a Serial Analysis of Gene Expression (SAGE) database with patient survival in SEER data [26]. Using an adapted approach, this study sought to identify prognostic and chemopredictive miRNAs by correlating their expression in a lung squamous cell carcinoma (LUSC) cohort (*n* = 57) [9] with clinical outcome of LUSC patients in the SEER-Medicare database (*n* = 33,897) as a function of tumor progression indicators combining T, N, and M cancer stage factors and tumor grade. The identified prognostic/chemopredictive miRNAs were then corroborated with the NCI-60 cell panel [27,28], NSCLC patient cohorts collected from Case Western Reserve University (CWRU) Comprehensive Cancer Center (*n* = 87) and West Virginia University Cancer Institute/Mary Babb Randolph Cancer Center (MBRCC; *n* = 69), and public NSCLC data including TCGA (*n* = 1016) and Cancer Cell Line Encyclopedia (CCLE; *n* = 117) [29]. We hypothesized that (1) correlating transcriptomic profiles in a clinical cohort with large-scale EMRs as a function of tumor progression indicators combined with rigorous external validation would identify prognostic and predictive biomarkers with consistent expression patterns in tumor progression, and (2) the biomarkers identified with this novel methodology would have potential prognostic implications in a large patient population. This approach is advantageous in taking into account the effects of surgical and radiological treatments in combination when assessing the role of miRNAs in mediating response to specific chemotherapy. Lastly, important miRNA-mediated regulatory pathways and proliferation networks in NSCLC were identified using public CRISPR-Cas9/RNAi screening data. The overall study scheme is delineated in Figure 1. Patient clinical characteristics in the studied cohorts are summarized in Table 1.

## 2. Results

### 2.1. Identification of Prognostic miRNAs for NSCLC

MiRNAs significantly (*p* < 0.05) associated with patient overall survival were identified from a squamous cell carcinoma cohort from Raponi et al. (*n* = 57) [9] using Cox modeling and Kaplan–Meier estimation. These miRNA expression profiles were correlated with disease-specific survival of squamous cell lung carcinoma patients in the SEER-Medicare database (*n* = 33,897). Specifically, the AJCC TNM classifications and tumor grade were used to partition patients into disjoint groups indicative of tumor progression, with the average survival being calculated for each group. Patients in the Raponi cohort [9] were partitioned in a similar manner. Linear regression was used to estimate the association between average miRNA expression in the Raponi cohort [9] and survival in SEER-Medicare patients. Prognostic miRNAs were identified by including SEER-Medicare patients without indication of having received chemotherapy in the analysis. In the evaluation of the results from the Raponi cohort [9], miRNAs with significant (*p* < 0.05) and concordant results on Cox modeling, Kaplan–Meier analysis, and linear regression in the correlation with SEER-Medicare data were identified as prognostic miRNAs (Figure 1 and Table 2). 

The identified prognostic miRNAs were further validated with NSCLC patient cohorts collected from the US hospitals MBRCC (*n =* 69) and CWRU (*n =* 87), as well as public TCGA data on lung adenocarcinoma and squamous cell lung carcinoma (TCGA-LUAD and TCGA-LUSC; *n* = 1016). The prognostic miRNAs with significant and concordant association with overall survival and/or recurrence-free survival in these external patient cohorts were selected (bolded in Table 2). Detailed results on external validation were provided in Appendix A.

### 2.2. Identification of Chemopredictive miRNAs for NSCLC

Similar analysis was performed to identify chemopredictive miRNAs in NSCLC. MiRNAs significantly associated with patient overall survival (*p* < 0.05; Cox modeling and Kaplan–Meier analysis) in the squamous cell carcinoma cohort from Raponi et al. (*n* = 57) [9] were correlated with disease-specific survival of squamous cell lung carcinoma patients in the SEER-Medicare database (*n* = 33,897). In this analysis, only patients treated with any chemotherapy or a specific chemotherapeutic agent were considered, and those who did not receive chemotherapy were excluded from the analysis. MiRNAs with significant (*p* < 0.05) and concordant results on Cox modeling, Kaplan–Meier analysis, and linear regression in the correlation with SEER-Medicare patients who received chemotherapy were identified as chemopredictive miRNAs (Figure 1, Table 2 and Table 3).

MiRNAs predictive of chemoresponse to cisplatin, carboplatin, etoposide, and paclitaxel identified in this population correlation study were validated with the drug activity profiles in the NCI-60 cell lines using linear regression. Cisplatin, carboplatin, docetaxel, erlotinib, etoposide, gefitinib, gemcitabine, pemetrexed, and vinorelbine in the CCLE data were included in the validation of chemosensitive (Table 3) and chemoresistant miRNAs (Table 4) for specific chemotherapy or the category of “any chemotherapy”. Significant differential expression (*p* < 0.05; two-sample *t*-tests) of miRNA in sensitive versus resistant NSCLC cell lines to specific drugs in the CCLE data were used as the evaluation criterion. The identified chemopredictive miRNAs were further validated with two patient cohorts from MBRCC (*n =* 69) and CWRU (*n =* 87) on the basis of the information of chemotherapy use in these patients. TCGA did not provide information on chemotherapy. Therefore, the complete TCGA NSCLC patient cohorts were used in the evaluation. A significant and concordant association with patient overall survival and/or recurrence-free survival (*p* < 0.05, Cox modeling and Kaplan–Meier analysis) was used to evaluate chemopredictive miRNAs. Predictive miRNAs with a confirmed result in the NCI-60 and CCLE cell panels and/or NSCLC patient cohorts from MBRCC, CWRU, or TCGA are in bold in Table 3 and Table 4. Detailed results on external validation are provided in Appendix A.

Due to the synergism and successful results of the combination of cisplatin–etoposide in treating small-cell lung cancer, long-term daily administration of oral etoposide in combination with cisplatin was used to treat NSCLC [30]. A systematic review showed that cisplatin–etoposide has comparable efficacy to carboplatin–paclitaxel when used with concurrent radiotherapy for patients with stage III unresectable NSCLC [31]. Paclitaxel, a tubulin-binding agent, is commonly used to treat NSCLC in combination with a platinum-based compound [32]. For cisplatin, carboplatin, paclitaxel, and etoposide, significant predictive miRNAs were also examined for interactions with genes known to be functionally involved in lung cancer or relevant cellular processes such as apoptosis, proliferation, cell-cycle regulation, or metastasis with Ingenuity Pathway Analysis (IPA) (Figure A1, Figure A2, Figure A3 and Figure A4, Appendix B, respectively).

### 2.3. Hsa-miR-142-3p/Hsa-miR-142 as a Good Prognostic and Chemosensitive Biomarker for NSCLC

Hsa-miR-142-3p was identified as a prognostic and chemosensitive biomarker from the population correlation with SEER-Medicare data (Table 2 and Table 3), and it was validated in all the studied datasets, including patient cohorts from MBRCC, CWRU, and TCGA, as well as drug activities in the cancer cell line panels NCI-60 and CCLE (Figure 2 and Figure 3).

The expression of hsa-miR-142-3p was positively correlated with disease-specific survival of SEER-Medicare patients who did not receive any chemotherapy (Table 2; Figure 2A). Hsa-miR-142-3p expression was positively correlated with chemosensitivity to paclitaxel in the NCI-60 cell lines (*p* = 0.0224, linear regression; Figure 2B). Squamous cell lung carcinoma patients in SEER-Medicare data who were correlated with a higher expression of hsa-miR-142-3p in the Raponi cohort [9] had a significantly longer survival in the treatment categories of any treatment plus paclitaxel (log-rank *p* < 0.0001; Figure 2C), as well as surgery plus paclitaxel (log-rank *p* = 0.0174; Figure 2D) in Kaplan–Meier analysis. These results indicate that a higher expression of hsa-miR-142-3p is associated with a lower risk of recurrence/metastasis in NSCLC patients and chemosensitivity to paclitaxel. These results were validated with NSCLC patient cohorts we collected from CWRU (Figure 2E,F) and MBRCC (Figure 2G–H). Patients from CWRU and MBRCC with a higher expression of hsa-miR-142-3p had a significantly longer survival time (log rank *p* < 0.05, Kaplan–Meier analysis) than those with a lower expression of hsa-miR-142-3p, in specific clinical settings of either receiving chemotherapy (Figure 2E,G) or not (Figure 2F,H). Chemotherapy in CWRU and MBRCC patient cohorts included cisplatin, carboplatin, paclitaxel, and pemetrexed. Due to the small sample size, CWRU and MBRCC cohorts were not split into specific LUAD or LUSC subtypes in the analysis. In this analysis, the miRNA expression was quantified with qPCR in the NCI-60 panel or microarray assays of patient tumors.

The above results were further validated in public RNA-sequencing data of NSCLC tumors from TCGA and cell lines from CCLE. When the gene-level expression was considered, both lung adenocarcinoma (Figure 3A,B) and squamous cell lung carcinoma patients (Figure 3C) with a higher expression of hsa-miR-142 had a significantly longer survival in Kaplan–Meier analysis. Next, all the isoforms of hsa-miR-142 were examined for their prognostic performance in TCGA NSCLC patients. A total of 14 isoforms of hsa-miR-142 had a hazard ratio (HR) less than 1 (*p* < 0.05, Cox modeling) in lung adenocarcinoma, indicating a positive association with patient survival (Figure 3D). A total of four isoforms of hsa-miR-142 had an HR greater than 1 (*p* < 0.05, Cox modeling) in lung adenocarcinoma, indicating a negative association with patient survival (Figure 3D). In squamous cell lung carcinoma, seven hsa-miR-142 isoforms had a positive association with survival and one isoform had a negative association with survival (*p* < 0.05, Cox modeling; Figure 3E). The expression of hsa-miR-142-3p was found to be positively associated with chemosensitivity to erlotinib in NSCLC cell lines, with a fold change of 0.45 with respect to underexpression in resistant versus sensitive lines (Figure 4). Gefitinib and erlotinib are widely used epidermal growth factor receptor (EGFR) tyrosine kinase inhibitors for treating advanced NSCLC with proven efficacy. A recent meta-analysis showed that gefitinib and erlotinib have comparable effects on patient survival, overall response rate, and disease control rate, with no considerable variation with regard to EGFR mutation status, ethnicity, line of treatment, and baseline brain metastasis status [33]. EGFR mutation in the studied NSCLC cell lines was provided in our previous study [34].

Together, these results indicate that hsa-miR-142-3p and the overall expression of hsa-miR-142 are positively associated with NSCLC survival. Hsa-miR-142-3p is a biomarker of chemosensitivity, including paclitaxel and erlotinib, in NSCLC patients and cell lines.

### 2.4. Hsa-miR-142-3p-Regulated Pathways and Functional Networks in NSCLC

The demonstrated clinical relevance of hsa-miR-142-3p substantiated further investigation of its molecular mechanisms. First, experimentally confirmed target genes of hsa-miR-142-3p were retrieved with TarBase [35]. A total of 842 target genes were included in the list. Analyzed with ToppGene [36], these hsa-miR-142-3p target genes were significantly enriched in gene families including zinc finger proteins, Kelch like/BTB domain-containing protein, RNA-binding motif containing protein, NADH ubiquinone oxidoreductase, death-inducing signaling complex, exportins, and super elongation complex (Figure 5A). Top significantly enriched pathways of hsa-miR-142-3p target genes were PDGFR-beta signaling pathway, E-cadherin signaling in the nascent adherens junction, ErbB1 downstream signaling, B-cell receptor signaling pathway, head and neck squamous cell carcinoma, insulin signaling, exercise-induced circadian regulation, VEGFA–VEGFR2 signaling pathway, p53 pathway feedback loops, and integrated breast cancer pathway (Figure 5B). These target genes were overrepresented in cytobands 19p12 (*p* < 1.7 × 10^−7^) and 8q13 (*p* < 1.1 × 10^−4^). Details are provided in Appendix A.

Next, molecular functions of 842 hsa-miR-142-3p target genes were investigated using public CRISPR-Cas9 [37] and RNAi [38] screening data in NSCLC cell lines. Nine experimentally confirmed target genes of hsa-miR-142-3p, *ARCN1* [39], *COPA* [40], *CSE1L* [41,42,43,44], *NUP205* [41], *RAN* [45], *RPS11* [40,42,43,44], *SF3B1* [39,42], *SNRNP200 (ASCC3L1)* [42], and *SNRPD1* [41], had a significant impact on proliferation in 100% of the NSCLC cell lines in CRISPR-Cas9 knockout (*n* = 78) and shRNA knockdown (*n* = 92). This hsa-miR-142-3p-regulated proliferation network in NSCLC is delineated in Figure 5C. Germline *COPA* mutations encoding the alpha-COP subunit of COPI impair ER–Golgi transport and cause hereditary autoimmune-mediated lung disease and arthritis [46]. Mutations of *ARCN1* [47], which encode the coatomer subunit delta of COPI [48], are linked to craniofacial syndrome due to COPI-Mediated transport defects [49]. CSE1L/CAS, an RNA-binding protein, is important in mitotic spindle checkpoint assuring genomic stability during cell division and is involved in proliferation and apoptosis [50]. TMEM209 and NUP205 protein interactions, stabilizing NUP205 and increasing the level of c-Myc in the nucleus, are critical drivers of lung cancer proliferation [51]. *RPS11* is involved in perturbed pathways in oncogene-induced senescence in human fibroblasts [52]. Hotspot mutations of *SF3B1* are associated with cancer and affect alternative splicing by promoting alternative branchpoint usage [53]. The affinity between DNA repair factors *ASCC2* and *ASCC3* is reduced by cancer mutations [54]. Depletion of SNRPE or SNRPD1 led to autophagy and a marked reduction of cell viability in breast, lung, and melanoma cancer cell lines, accompanied by a deregulation of the mTOR pathway [55]. *SNRNP200*, *ARCN1*, *COPA*, and *SF3B1* had a significant negative correlation with the overall expression of hsa-miR-142 (*p* < 0.05; Pearson’s correlation) in the combined TCGA-LUAD and TCGA-LUSC data [56]. Among these hsa-miR-142-3p target genes involved in NSCLC proliferation, *NUP205*, *RAN*, *CSE1L*, *SNRPD1*, and *RPS11* had a significant positive correlation with overall expression of hsa-miR-142 (*p* < 0.05; Pearson’s correlation) in the combined TCGA-LUAD and TCGA-LUSC patients. Recent pan-cancer analysis of TCGA data found that positive miRNA–gene correlations are surprisingly prevalent and consistent across cancer types [57], which is consistent with the observation here. These genes were associated with chemosensitivity or chemoresistance to cisplatin, erlotinib, gefitinib, paclitaxel, and pemetrexed in CCLE NSCLC cell lines (Table 5; Appendix A). Further investigation in patient tumor tissues using qRT-PCR, ELISA, and/or immunohistochemistry is warranted to substantiate the clinical relevance of these genes in NSCLC treatment. 

Gene Set Enrichment Analysis (GSEA) [58,59] was used to identify significant pathways and gene sets of hsa-miR-142-3p target genes in short- vs. long-term survival using mRNA expression data of three patient cohorts: GSE28582 [60,61], GSE81089 [62], and TCGA (combined TCGA-LUAD and TCGA-LUSC) [56]. Patients who survived shorter than 20 months after treatment were defined as short-term survivors, and those who survived longer than 58 months after treatment were defined as long-term survivors. Six pathways/gene sets were significantly enriched in short-term survivors in all three patient cohorts (Figure 5D), including genes downregulated in SaOS-2 cells (osteosarcoma) upon knockdown of *YY1* by RNAi (DE_YY1_TARGETS_DN), process of lamellipodium organization (GOBP_LAMELLIPODIUM_ORGANIZATION), genes upregulated in livers injected with *IL6*: *SOCS3* knockout versus wild-type (GSE369_SOCS3_KO_VS_WT_LIVER_POST_IL6_INJECTION_UP), EPH–Ephrin signaling (REACTOME_EPH_EPHRIN_SIGNALING), RHO GTPase effectors (REACTOME_RHO_GTPASE_EFFECTORS), and genes downregulated in primary tissue culture of epidermal keratinocytes after UVB irradiation (TAKAO_RESPONSE_TO_UVB_RADIATION_DN). 

Genes involved in these pathways and gene sets that were overrepresented in NSCLC short-term survivors are shown in Figure 5D. The mRNA expression of *ARPC1B*, *KTN1*, and *WASL* had a significant negative correlation with hsa-miR-142-3p in CCLE NSCLC cell lines (*p* < 0.03, Pearson’s correlation). *KTN1* and *WASL* also had a significant negative correlation with hsa-miR-142 in the combined TCGA-LUAD and TCGA-LUSC data (*p* < 5.1 × 10^−^^6^, Pearson’s correlation). A total of 15 target genes had a significant negative correlation and 21 target genes had a significant positive correlation with hsa-miR-142 in the combined TCGA-LUAD and TCGA-LUSC data (*p* < 0.02, Pearson’s correlation, Appendix A). 

Functional assessment of these genes showed that six genes had a significant impact on cell proliferation in at least 61% of the NSCLC cell lines in both CRSIPR-Cas9 (*n* = 78) and RNAi (*n* = 92) screening: *PRC1*, *PSMA4*, *NAA15*, *HNRNPC*, *RINT1 (RAD50)*, and *XPO1* (Appendix A). *PRC1* contributes to tumorigenesis of lung adenocarcinoma and plays a key role in the activation of the Wnt/β-catenin signaling pathway [63]. A haplotype-based association analysis found that *PSMA4* is a strong candidate mediator of lung cancer cell growth, and may directly affect lung cancer susceptibility through its modulation of cell proliferation and apoptosis [64]. Truncating variants in *NAA15* are associated with intellectual disability and congenital abnormalities [65]. High *HNRNPC* mRNA and protein expression is significantly related to poor overall survival in patients with lung adenocarcinoma [66]. Rare mutations in *RINT1 (RAD50)* are predisposition risk factors of breast and Lynch syndrome-spectrum cancers [67]. The *MRE11–RAD50–NBS1* complex is essential in DNA damage repair and tumorigenesis and is a promising target in cancer treatment [68]. *XPO1* inhibitors are promising therapeutic strategies in *KRAS*-mutant lung cancer [69]. The identified hsa-miR-142-3p-regulated pathways and networks in short-survival NSCLC provide novel and important insights into disease mechanisms and potential intervention strategies.

## 3. Discussion

Lung cancer is difficult to manage in clinics due to its complex somatic mutations and etiology, and it remains the leading cause of cancer death in the US and worldwide for both men and women. To date, there are no effective molecular biomarkers to recommend optimal treatment selection for individual patients, including specific chemotherapy, use of immunotherapy, and combination with radiation therapy for NSCLC patients with all stages and subtypes. MiRNAs are promising biomarkers in diagnostics and prognostics due to their greater stability in prepared tissue samples including formalin fixation [16] compared with mRNA [14,15], as well as their presence in circulating plasma [12,13]. Studies on miRNAs could also lead to the development of novel intervention strategies by revealing post-transcriptional regulatory pathways and networks [17,18,19].

This study sought to identify prognostic and chemopredictive miRNAs to improve treatment for NSCLC. The patient cohorts we collected from US hospitals (MBRCC and CWRU) had limited sample size. The small sample size made it infeasible to correlate miRNA expression to the clinical outcome of specific chemotherapy (i.e., cisplatin, carboplatin, paclitaxel, pemetrexed, etc.) in a particular NSCLC subtype (i.e., adenocarcinoma or squamous cell carcinoma). The public TCGA data did not provide chemotherapy information. The drug activity data in the NCI-60 and CCLE panels are in vitro and do not always represent the results in patients. 

To overcome these limitations, this study presents a novel methodology for identifying prognostic and chemopredictive biomarkers with potential to be applied in large patient populations by integrating SEER-Medicare data. There has been reported success in identifying chemopredictive genes by correlating mRNA expression profiles in solid tumors of the SAGE database with patient survival in the SEER data [26]. In the study by Stein et al. [26], genes differentially expressed between solid tumors and cell lines were first selected from the SAGE database without being adjusted for multiple testing. To control false discovery, gene expression in solid tumors of the SAGE database was correlated with a 5-year survival of patients with a distant disease in the SEER data by different tumor types. Stein et al. [26] used the SEER 5 year survival data as a surrogate for chemosensitivity, acknowledging that factors other than chemosensitivity also influence patient survival. Here, since we focused on NSCLC, the approach presented by Stein et al. [26] was adapted in the following way: a composite tumor progression indicator based on AJCC TNM cancer stage and tumor grade was used as a surrogate to correlate miRNA expression with SEER-Medicare patient survival, recognizing that the cancer stage is a strong indicator of NSCLC survival. Specifically, the AJCC TNM cancer staging classifications and tumor grade (G) were used to partition SEER-Medicare patients into disjoint groups (i.e., T = 1; N = 0; M = 0; G = 1) indicative of tumor progression, with the average survival being calculated for each group. Patients in the training clinical cohort from Raponi et al. [9] were partitioned in a similar manner. Linear regression was used to estimate the association between average miRNA expression (per group) in the training clinical cohort from Raponi et al. [9] and the corresponding average survival in SEER-Medicare patients. The results were compared with those from independent clinical validation cohorts in the public domain, including our cohorts upon publication. Drug activities of commonly used chemotherapy, together with miRNA and mRNA profiles in the NCI-60 and CCLE cell lines, were used to corroborate the results on chemoresponse prediction. Similar to Stein et al. [26], prognostic and chemopredictive miRNAs were first selected using univariate Cox model and Kaplan–Meier analysis in the training clinical cohort from Raponi et al. [9] without multiple testing. The false discovery was controlled (1) by population correlation with SEER-Medicare data with linear regression and corroboration with Kaplan–Meier analysis, and (2) with additional external validation using our collected patient cohorts, as well as TCGA, NCI-60, and CCLE data.

The results demonstrate that, according to similarities in tumor progression, extrapolation of miRNA expression from smaller cohorts to larger population-based data can serve as an additional confirmatory tool where novel cohorts containing tens of thousands of patients with matched clinical outcomes and genome-scale transcriptomic profiles are unavailable. This approach, when combined with rigorous external validation, can identify miRNAs with consistent expression patterns in tumor progression, with potential prognostic and predictive implications in large patient populations. 

Specifically, it was shown that multiple miRNAs were selected as strong predictors of chemoresponse through analysis of disease-specific survival in cohorts with similar treatment strategies. The miRNAs validated with the external patient cohorts and public data have the potential to be used in clinical practice for prognosis and the selection of chemotherapy in NSCLC treatment. Among the selected miRNAs, hsa-miR-142-3p was validated as a good prognosis and chemosensitivity biomarker in all the studied NSCLC patient cohorts (*n* = 1172) and drug activities in the NCI-60 panel and CCLE NSCLC cell lines (*n* = 117), with promising results for clinical utility upon further prospective evaluation. In the functional assessment of hsa-miR-142-3p target genes using public in vitro CRISPR-Cas9/RNAi screening data, the hsa-miR-142-3p-mediated proliferation network in NSCLC was identified. Furthermore, hsa-miR-142-3p-regulated pathways and functional networks in NSCLC with poor prognosis (short-term survival) were elucidated. Functioning as a tumor suppressor, miR-142-3p represses TGF-β-induced growth inhibition [70] and inhibits lung cancer progression through repressing β-catenin expression [71]. This study provides novel insights into hsa-miR-142-3p-mediated cancer cell proliferation and tumor progression in NSCLC for the development of therapeutic strategies.

Several selected prognostic and chemopredictive miRNAs are consistent with their reported roles in lung cancer. MiR-29 was selected as a chemosensitive biomarker. MiR-29 family reverts abnormal methylation in lung cancer by targeting DNA methyltransferases 3A and 3B (*DNMT3A* and *DNMT3B*), induces re-expression of methylation-silenced tumor suppressor genes, such as *FHIT* and *WWOX*, and inhibits tumorigenicity in vitro and in vivo [72]. MiR-192 was selected as a chemoresistant biomarker in this study, consistent with its role in enhancing chemoresistance and invasiveness of LUSC [73]. MiR-134 has ambiguous roles in human cancer [74]. MiR-134 was selected as a good prognosis and chemosensitive biomarker in this study. MiR-134 inhibits NSCLC growth through mechanisms including targeting EGFR [75], downregulating oncogene *CCND1* [76], and inhibiting epithelial-to-mesenchymal transition (EMT) by targeting *FOXM1* [77]. On the other hand, miR-134-5p was reported to promote metastasis and chemoresistance by targeting *DAB2* in stage I lung adenocarcinoma [78]. Recent deep RNA-sequencing studies revealed that miRNA isoforms are biologically relevant and functionally cooperative partners of canonical miRNAs in post-transcriptional regulation [79]. Different isoforms of the same miRNA may have markedly different prognostic implications and functions, as evidenced in Figure 3D,E. In this study, miR-134 had a positive correlation with LUSC survival in the SEER-Medicare data and TCGA-LUSC. In contrast, miR-134 had a negative association with survival in TCGA-LUAD and was overexpressed in NSCLC cell lines resistant to cisplatin and docetaxel in CCLE GDSC2 data (Appendix A). Docetaxel offers clinical benefits as a second-line treatment of NSCLC in patients previously treated with platinum-based chemotherapy [80]. It was recently reported that the combination of pembrolizumab (anti-PD1 immunotherapy) plus docetaxel was well tolerated and substantially improved progression-free survival and overall response rate in patients with advanced NSCLC after platinum-based chemotherapy, including patients with EGFR variations [81]. These results are consistent with reported ambiguous roles of miR-134, which are possibly due to the different studied isoforms and NSCLC subtypes. More miRNAs selected using the population correlation approach are anticipated to be validated with new upcoming transcriptomic profiling of more patients and functional studies of miRNAs. 

The methodology presented in this study is general and could be applied to other types of molecular profiles and cancer types. By design, the SEER registry represents a more demographically and clinically diverse group of patients when compared to cohorts limited to a specific geographic area or healthcare system. Nevertheless, the use of administrative data also has some limitations. Although the SEER data are in general highly accurate, they are not fully inclusive of all treatments a patient may receive. Treatments not covered by Medicare or covered by other forms of insurance would not appear in the database. Additionally, eligibility requirements for Medicare coverage artificially limit the patient sample to those over the age of 65, with notable exceptions such as eligibility due to disability benefits. As lung cancer occurs primarily in older populations, this effect is limited relative to other cancer types which are common in comparatively younger populations.

## 4. Materials and Methods

### 4.1. Patient Cohorts Used in Correlation with SEER-Medicare Data

A cohort of 57 squamous cell carcinoma patients originally published by Raponi et al. [9] was used to correlate with SEER-Medicare data. Included in this cohort are expression data on 328 human miRNA quantified by MirVana miRNA Bioarrays, with corresponding follow-up information on survival time and status, tumor grade, and AJCC tumor T, N, and M markers. Tumor grade for this cohort was converted from a descriptive measure of tumor differentiation to numerical grade to match descriptors used in SEER-Medicare data. This dataset is available from the NCBI Gene Expression Omnibus (GEO) with accession number GSE 16025. (https://www.ncbi.nlm.nih.gov/geo/query/acc.cgi?acc=GSE16025, accessed on 16 July 2021).

The linked SEER-Medicare database [82] combines the clinical, demographic, pathological, and survival information from the NCI SEER registry system with claims data for individual patients found in the Medicare claims database. Briefly, patients in the SEER portion were linked according to identifying information such as social security number, census tract, age, and other identifying criteria to records found in the Medicare claims database [83]. Criteria for inclusion in this study were a diagnosis of squamous cell carcinoma of the lung or bronchus between 1991 and 2005, as well as valid information on survival time and status, tumor grade, and AJCC T, N, and M markers. A total of 33,897 patients fit these criteria, with this cohort being stratified according to treatment modality in subsequent analyses. Patient demographics for this group are detailed in Table 1. The T, N, and M markers for patients diagnosed prior to 2004 were derived from EOD10 coding, where possible, and according to established conversion algorithms [25]. Four specific chemotherapeutic agents, cisplatin, carboplatin, paclitaxel, and etoposide were considered. The administration of chemotherapy was determined through the use of Healthcare Common Procedure Coding System (HCPCS) codes [82]. First, the use of chemotherapy was determined by searching individual patient claims histories for entries with an HCPCS code corresponding to the agent in question. The ICD-9 diagnosis codes for these records were then checked to ensure that the agent was administered for the treatment of lung cancer. Curative surgery and radiation therapy were determined using variables in the SEER portion of the data. These variables were used to stratify patients into a group receiving any surgical procedure but not preoperative radiation, a group with any type of radiation but not surgery, a group with both surgery and radiation, and a group with any combination of treatments. Survival estimates were represented as disease-specific survival for the SEER-Medicare cohort and overall survival for the clinical cohort from Raponi et al. [9]. 

### 4.2. Validation NSCLC Patient Cohorts

Three NSCLC patient cohorts were used in the validation. The first validation patient cohort contained 69 NSCLC patient samples collected from MBRCC and Cooperative Human Tissue Network (CHTN) operated by the NCI. MiRNA expression profiles were generated with Sanger 15 (Ocean Ridge Biosciences, Deerfield Beach, FL, USA). These data are available from the NCBI GEO with accession number GSE32524. The second validation patient cohort contained 87 NSCLC samples collected from CWRU. miRNA expression profiles were generated with Sanger 15 (Ocean Ridge Biosciences, Deerfield Beach, FL, USA). These data are available from the NCBI GEO with accession number GSE31275. The third validation cohort was from TCGA. MiRNA expression data of TCGA lung adenocarcinoma (TCGA-LUAD) and TCGA lung squamous cell carcinoma (TCGA-LUSC) are available from LinkedOmics [56] (http://linkedomics.org/; accessed on 28 April 2021). Gene-level data of the Illumina GenomeAnalyzer platform and Illumina HiSeq platform (Illumina, San Diego, CA, USA), and miRNA isoform-level data were included in the validation. The miRNA expression values were log-transformed.

Illumina HiSeq platform RNA-Seq mRNA expression data of TCGA-LUAD and TCGA-LUSC were also downloaded from LinkedOmics. The mRNA expression values were log-transformed. TCGA-LUAD and TCGA-LUSC were combined in the correlation calculation of miRNA and mRNA expression in the same patients.

### 4.3. NCI-60 Cellular Data

The NCI-60 cellular data were used to evaluate chemopredictive miRNAs [27,28]. This dataset contains expression levels of 209 miRNAs measured with quantitative PCR across 59 cancer-derived cell lines of diverse tissue origin. The data also contain the drug activity at three clinically relevant endpoints: 50% growth inhibition (GI_50_), Total growth inhibition (TGI), and 50% lethal concentration (LC_50_), used to refine the set of significant miRNAs and assess meaningful biological context.

### 4.4. Cancer Cell Line Encyclopedia (CCLE)

MiRNA and mRNA expression data for CCLE were downloaded from DepMap 20Q2 (https://figshare.com/articles/dataset/DepMap_20Q2_Public/12280541, accessed on 1 April 2021) [84]. Gene expression data were obtained from the CCLE data portal (https://data.broadinstitute.org/ccle/CCLE_RNAseq_081117.rpkm.gct, accessed on 1 April 2021). RNA-seq data were quantified using the GTEx pipelines [85]. A total of 117 NSCLC cell lines were included in this analysis.

### 4.5. PRISM Drug Response in CCLE

The growth-inhibitory activity of 4518 drugs was quantified in 578 human cancer cell lines using the PRISM molecular barcoding and multiplexed screening method [86]. The PRISM repurposing dataset is available at the Cancer Dependency Map portal (https://depmap.org/portal/download/, accessed on 1 April 2021). Drug responses of nine commonly used chemotherapeutic regimens in treating NSCLC were included in this study: carboplatin, cisplatin, paclitaxel, docetaxel, gemcitabine, vinorelbine, etoposide, gefitinib, and erlotinib. For each drug, cell lines with IC_50_ or EC_50_ values higher than the maximum dose were defined as resistant; cell lines with IC_50_ or EC_50_ value lower than the minimum dose were defined as sensitive. The remaining cell lines were divided into groups of resistant, sensitive, or partial response by using the mean ± 0.5 standard deviation (SD) of the IC_50_, ln(IC_50_), EC_50_, or ln(EC_50_) values [87,88]. Cell lines with an IC_50_, ln(IC_50_), EC_50_, or ln(EC_50_) value greater than the mean + 0.5 SD were defined as resistant to the drug. Cell lines with an IC_50_, ln(IC_50_), EC_50_, or ln(EC_50_) value less than the mean − 0.5 SD were defined as sensitive to the drug, and those with an IC_50_, ln(IC_50_), EC_50_, or ln(EC_50_) value between the mean + 0.5 SD and the mean − 0.5 SD were defined as having a partial response to the drug. This categorization corresponds to the RECIST 1.1 system (i.e., complete response, partial response, and stable disease/disease progression) in evaluating chemotherapeutic response in solid tumors [89].

### 4.6. Genomics of Drug Sensitivity in Cancer (GDSC1/2)

Drug screening data were downloaded from Genomics of Drug Sensitivity in Cancer (GDSC) Project [90] (https://www.cancerrxgene.org/downloads/bulk_download, accessed on 15 April 2021). The GDSC Project screened more than 1000 genetically characterized human cancer cell lines with a wide range of anticancer therapeutic agents. Among the commonly used chemotherapeutic regimens in treating NSCLC drugs, nine were found in GDSC1, namely, cisplatin, docetaxel, erlotinib, etoposide, gefitinib, gemcitabine, pemetrexed, and vinorelbine, while seven were found in GDSC2, namely, cisplatin, docetaxel, erlotinib, gefitinib, gemcitabine, paclitaxel, and vinorelbine. For each drug, cell lines were defined as resistant, sensitive, or partial response by using the mean ± 0.5 SD of the IC_50_ values as described above.

### 4.7. Identification of Prognostic miRNAs by Correlating with SEER-Medicare Data

Patients in the Raponi cohort [9] and SEER-Medicare sets were assigned to disjoint tumor progression groups according to unique combinations of tumor grade and T, N, and M markers. This tumor progression group membership served as a link between the two patient sets in subsequent analyses. In total, there were 16 groups common between the clinical and population datasets, including tumor grades 1 through 3, tumor T 1 through 4, and tumor N 0 through 2. There were no metastatic or grade 4 groups common between the two datasets. A leverage analysis was performed to test for undue influence from any one group. None of the groups were seen to have a disproportionate effect on model coefficients or significance; therefore, all 16 groups were included.

The set of prognostic miRNAs was determined by selecting miRNAs which showed a significant association with survival in the original clinical cohort from Raponi et al. [9], before validating these miRNAs in the SEER-Medicare population cohort. Cox modeling and Kaplan–Meier estimation were used to assess the association between expression and survival in the original clinical cohort from Raponi et al. [9]. Cox model coefficients and *p*-values were estimated for each miRNA in independent models. In the Kaplan–Meier analysis, cutoff values ranging from the 5% to 95% quantiles of miRNA expression were used to split the patients into overexpression and underexpression groups for survival analysis. The degree of separation between the resulting survival curves was estimated as a log-rank *p*-value, with a *p*-value less than 0.05 being deemed as significant separation between the prognostic groups.

Validation of the results of the clinical analysis on the SEER-Medicare population data was done using linear regression, Cox modeling, and Kaplan–Meier estimation. Multiple methods of assessment were chosen due to limitations associated with using a single measure [91]. The linear regressions used the average miRNA expression from the clinical set and average disease-specific survival from the SEER-Medicare population set for each tumor progression group. Average survival in the SEER-Medicare population cohort was calculated as a function of the area under the curve produced by Kaplan–Meier estimation. Each miRNA was evaluated in each of the four surgical and radiological treatment modalities. In order to enter the final set of prognostic miRNAs, each miRNA found to be significant in the original Raponi cohort [9] had to show a concordant and significant association with average survival in the SEER-Medicare population cohort. Individual patients in the SEER-Medicare population cohort were then assigned the average miRNA expression for that progression group from the clinical cohort from Raponi et al. [9], and the Cox and Kaplan–Meier models were re-evaluated. Any miRNAs which failed to achieve a significant and concordant association with survival in either the Cox or the Kaplan–Meier model were removed.

### 4.8. Identification of Chemopredictive miRNA Using Linked SEER-Medicare Data

In order to select for miRNAs which were predictive of chemoresponse, as represented by improved or diminished disease-specific survival, linear regression was again used to estimate the association between average miRNA expression in the clinical cohort from Raponi et al. [9] and disease-specific survival in the SEER-Medicare population cohort by tumor progression group. Next, each patient in the SEER-Medicare population set was assigned the average expression for each miRNA corresponding to the same tumor progression group in the clinical cohort from Raponi et al. [9]. These combined patient expression and survival data were then used to estimate a Cox proportional hazards model. A Kaplan–Meier model was also estimated and assessed on the log-rank *p*-values. Significance and concordance on the linear model and one of either the Cox or the Kaplan–Meier models represented sufficient evidence in the population analysis. Selection of a particular miRNA as a prognostic marker was not a requirement for selection as a predictive marker. 

### 4.9. Validation of Chemopredictive miRNAs Using the NCI-60 Anticancer Screen

The miRNAs which were significant in the chemopredictive correlation analyses were then compared to data from the NCI-60 anticancer screen [27,92,93]. Linear regression was used to estimate the association between expression on a specific miRNA marker in each cell line and drug activity in the same cell line for each of the previously described measures. Cell lines which did not have informative values at any dosage were removed from the analysis. Significance and concordance on any one of the three drug activity measures (GI_50_, LC_50_, and TGI) was considered in support of meaningful biological context.

### 4.10. Validation of Prognostic and Predictive miRNAs Using Independent Patient Cohorts and CCLE

The prognostic and chemopredictive miRNA biomarkers identified in the previous analysis were further validated with three external patient cohorts, MBRCC/CHTN, CWRU, and TCGA. Each miRNA was evaluated with the Cox proportional hazard model and Kaplan–Meier analysis. Patient subgroups receiving different kinds of treatments were analyzed separately. MiRNAs that have a significant association with overall survival and/or recurrence-free survival in treatment subgroups were selected. Furthermore, chemopredictive miRNAs were corroborated with their differential expression in sensitive versus resistant NSCLC cell lines to the studied drugs using CCLE PRISM and GDSC1/2 datasets.

### 4.11. Analysis of Target Genes of Hsa-miR-142-3p

A list of target genes featuring experimentally confirmed interactions with hsa-miR-142-3p were obtained from TarBase [35] v7.0 (http://snf-515788.vm.okeanos.grnet.gr/index.php, accessed on 5 May 2021) and v8.0 (https://carolina.imis.athena-innovation.gr/diana_tools/web/index.php?r=, accessed on 5 May 2021). A total of 842 target genes were included in the list. ToppFun of ToppGene suite [36] was used to detect functional enrichment of the hsa-miR-142-3p target gene list retrieved from TarBase. A total of 802 out of 842 target genes were available in ToppFun. The ToppFun tool used the false discovery rate (FDR) multiple correction method with enrichment significance cutoff level of 0.05 in the gene list enrichment analysis. Gene Set Enrichment Analysis (GSEA) software (version 4.1.0) [58,59] was used to identify significant pathways and gene sets of hsa-miR-142-3p target genes in short- vs. long-term survival using mRNA expression data of three patient cohorts: GSE28582 [60,61], GSE81089 [62], and TCGA (the combined TCGA-LUAD and TCGA-LUSC) [56]. In GSEA, the FDR *q*-value was used to select candidate gene sets. An FDR *q*-value < 0.05 indicates statistical significance. 

### 4.12. CRISPR-Cas9 Assays

Gene knockout effects in CCLE using CRISPR-Cas9 screens were quantified in Project Achilles [37,94]. The data were obtained from DepMap 20Q2 (https://figshare.com/articles/dataset/DepMap_20Q2_Public/12280541, accessed on 1 April 2021) [84]. The CRISPR-Cas9 data were processed with the CERES method [37]. Gene effects in each cell line were normalized such that the median nonessential gene knockout effect was 0 and the median essential gene knockout effect was −1. A gene is defined as an essential gene if it is essential to the cell growth in each line; otherwise, it is defined as a nonessential gene. A dependence score of −0.5 is indicative of a significant effect in CRISPR-Cas9 knockout. There were 78 NSCLC cell lines with genome-scale CRISPR-Cas9 knockout results. 

### 4.13. RNAi Functional Assays

Genome-scale RNAi screening data in CCLE were obtained from Project Achilles [38] (https://depmap.org/R2-D2/, accessed on 1 April 2021). The DEMETER2 method [38] was used to estimate average gene dependency scores in each cell line for short hairpin RNA (shRNA) libraries. Gene dependency scores were standardized with DEMETER2 such that the median of the across-cell-line average dependency scores of the positive control gene set was −1 and that of the negative control gene set was 0. A dependence score of −0.5 is indicative of a significant effect in shRNA knockdown. There were 92 NSCLC cell lines with genome-scale RNAi screening results normalized with DEMTER2.

### 4.14. Ingenuity Pathway Analysis

For each chemotherapeutic agent, the final set of significant miRNAs were also examined for interactions with molecular species known to play a role in lung cancer or relevant cellular processes such as apoptosis, proliferation, cell-cycle regulation, or metastasis with Qiagen Ingenuity Pathway Analysis (IPA) (Ingenuity^®^ Systems, Qiagen, Hilden, Germany) [95]. In short, IPA is a functional pathway analysis tool incorporating genes, cellular species such as proteins, and chemical compounds with data on their interactions and involvement in diseases derived from scholarly publications. Using these data, it is possible to map interactions between biomarkers on any given criterion. The list of significant miRNAs from each treatment was used to create networks on the basis of experimentally validated interactions between miRNA and molecular components with known biological function. These networks were created using the Core Analysis feature of IPA. The Core Analysis compares the set of miRNA markers with molecules with known roles in human disease in order to select a set of networks in which the interactions between the miRNA set and IPA-defined functional set are statistically over-represented. IPA was also used to retrieve reported interactions among hsa-miR-142-3p target genes.

### 4.15. Statistical Analysis

Statistical analysis was performed using Rstudio version 1.4.1106 [96]. Differential gene expression between two groups was evaluated with Student’s *t*-tests, and a two-sided *p*-value < 0.05 was considered statistically significant. Survival analysis was performed using Kaplan–Meier analysis with the survival package in R. Log-rank tests were used to assess the difference in survival probability from different groups in Kaplan–Meier analyses. Pearson’s correlation test was used to find the relationship between two variables.

## 5. Conclusions

This study developed a novel methodology to correlate genome-scale miRNA expression profiles in a LUSC clinical cohort with SEER-Medicare LUSC patients as a function of composite tumor progression indicators of T, N, and M cancer stage and tumor grade. The selected prognostic and chemopredictive miRNAs were extensively validated with miRNA profiles of NSCLC patient cohorts collected from US hospitals and public consortia including NCI-60, TCGA, and CCLE. Among the identified miRNA biomarkers, hsa-miR-142-3p was associated with good prognosis and chemosensitivity in NSCLC in all studied datasets. The functional assessment of hsa-miR-142-3p target genes using CRISPR-Cas9/RNAi screening data identified genes with a significant impact on proliferation in 100% of the tested NSCLC cell lines. Hsa-miR-142-3p-mediated pathways and functional networks in short-term survival NSCLC were elucidated. These results shed light on important molecular disease mechanisms underlying NSCLC and have the potential to develop novel therapeutic strategies. Overall, the approach integrating SEER-Medicare data combined with comprehensive external validation can identify miRNAs with consistent expression patterns in tumor progression, with potential implications for prognosis and prediction of chemoresponse in large NSCLC patient populations. This approach can be generalized to other molecular profiles and cancer types.

## Figures and Tables

**Figure 1 ijms-22-07658-f001:**
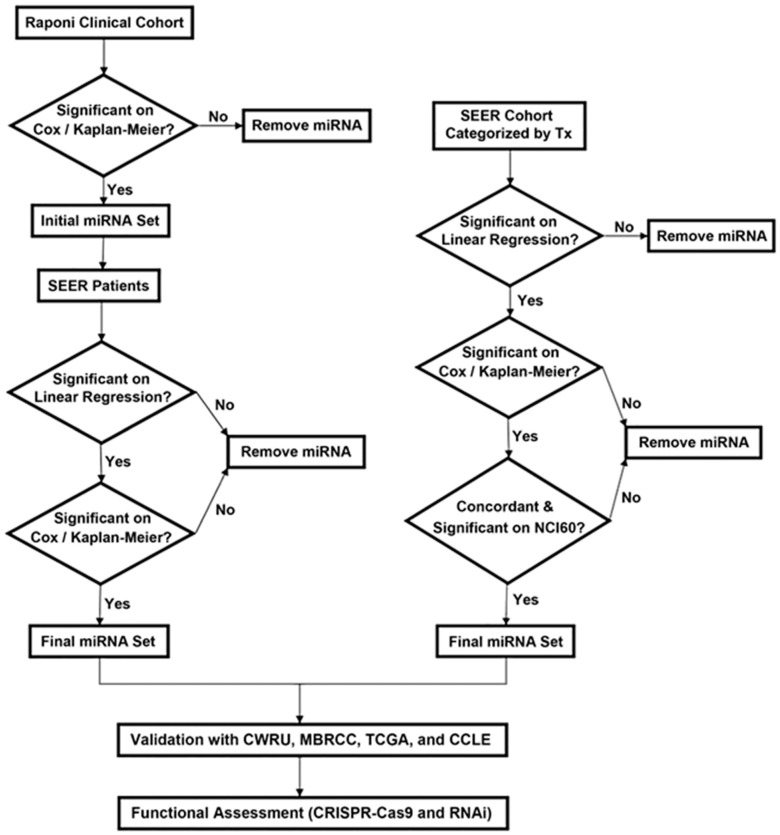
The overall study scheme. The arrows indicate the study flow. Tx: treatment. CWRU: Case Western Reserve University. MBRCC: Mary Babb Randolph Cancer Center.

**Figure 2 ijms-22-07658-f002:**
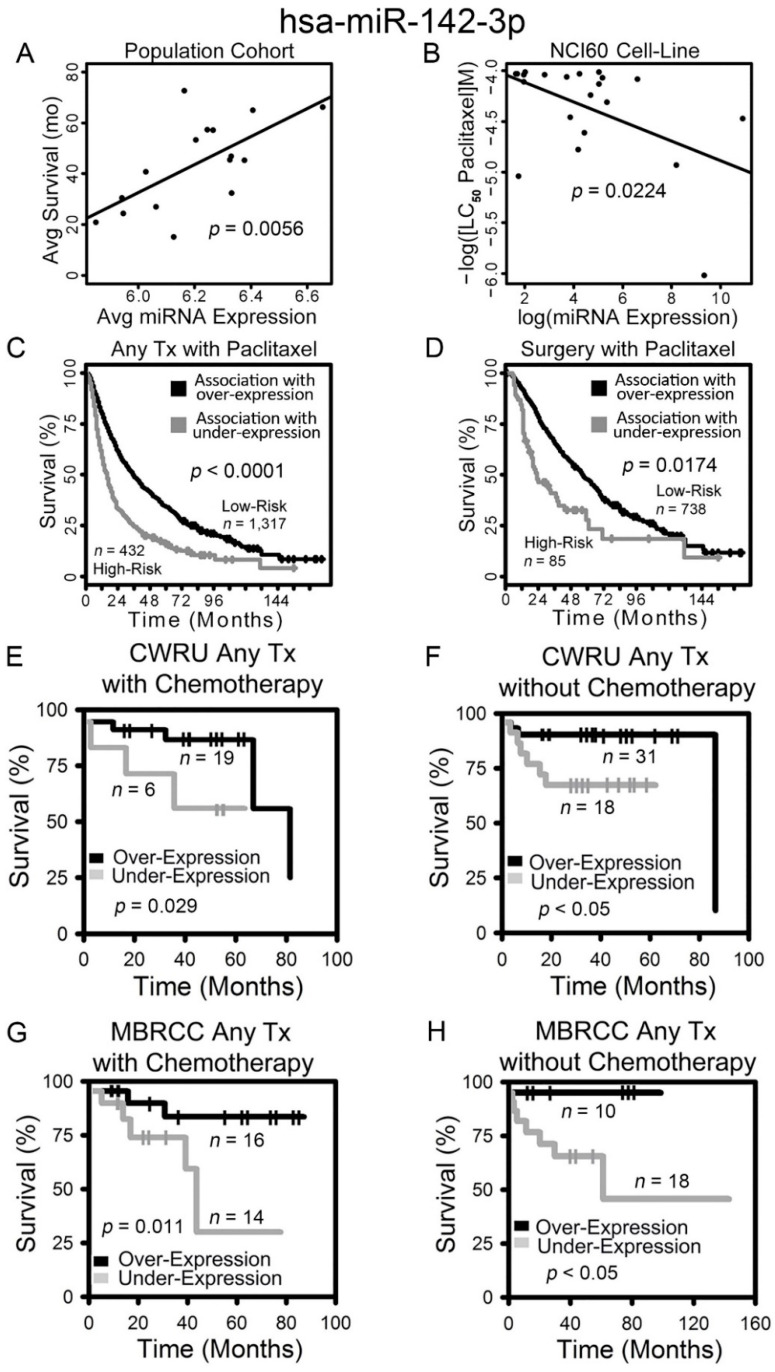
Overexpression of hsa-miR-142-3p correlates with prolonged survival and chemosensitivity in non-small cell lung cancer (NSCLC) patients. Correlation between hsa-miR-142-3p expression and survival in SEER-Medicare lung squamous cell carcinoma (LUSC) patient population (**A**), and the dosage required for lethal concentration (LC_50_) of paclitaxel in the NCI-60 (**B**). Kaplan–Meier analyses of patient stratification based on hsa-miR-142-3p expression in all SEER-Medicare LUSC patients treated with paclitaxel (**C**) and patients receiving only surgery and paclitaxel (**D**), CWRU patients receiving chemotherapy (**E**) or without chemotherapy (**F**), and MBRCC patients receiving chemotherapy (**G**) or without chemotherapy (**H**). Log-rank tests were used to assess the statistical significance.

**Figure 3 ijms-22-07658-f003:**
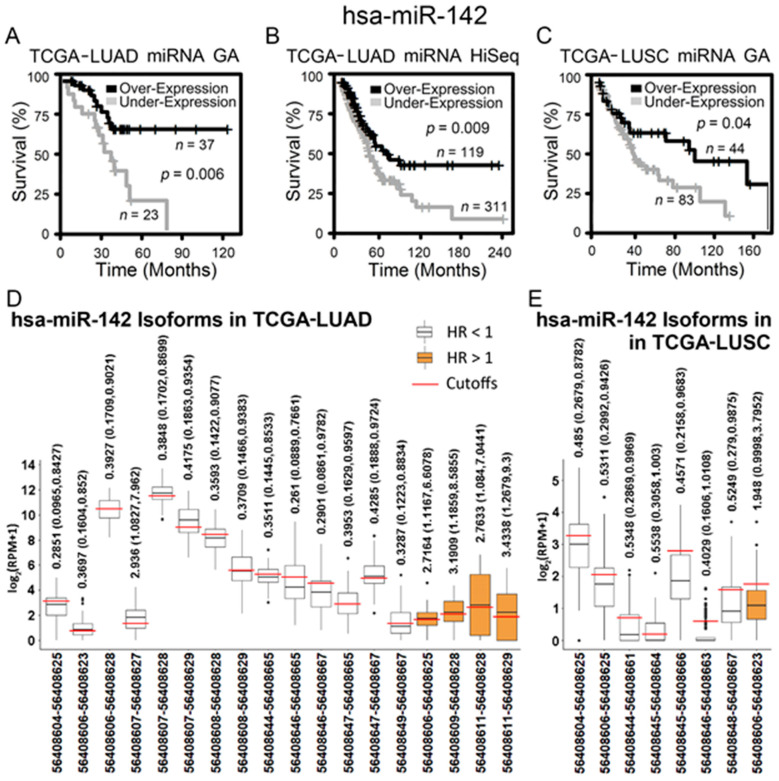
RNA-sequencing data of hsa-miR-142 and its isoforms in The Cancer Genome Atlas lung adenocarcinoma (TCGA-LUAD) and lung squamous cell carcinoma (TCGA-LUSC) patient cohorts. (**A**–**C**) Kaplan–Meier analysis of patient tumors grouped by hsa-miR-142 differential expression in TCGA-LUAD and TCGA-LUSC. When hsa-miR-142 was overexpressed, patients survived for a significantly longer period of time. The cutoff points of hsa-miR-142 overexpression and underexpression were: (**A**) miRNA cutoff = 12.5 in TCGA-LUAD GenomeAnalyzer (GA), (**B**) miRNA cutoff = 11.7 in TCGA-LUAD HiSeq, and (**C**) miRNA cutoff = 13.1 in TCGA-LUSC GenomeAnalyzer (GA). (**D**,**E**) The expression of hsa-miR-142 isoforms (with positions in chromosome 17 according to Human Genome 19 (hg19)) that had a significant hazard ratio (HR) in patient groups of overexpression versus underexpression in TCGA-LUAD (**D**) and TCGA-LUSC (**E**). Each isoform had a specific expression cutoff point to generate the corresponding HR shown in the figure.

**Figure 4 ijms-22-07658-f004:**
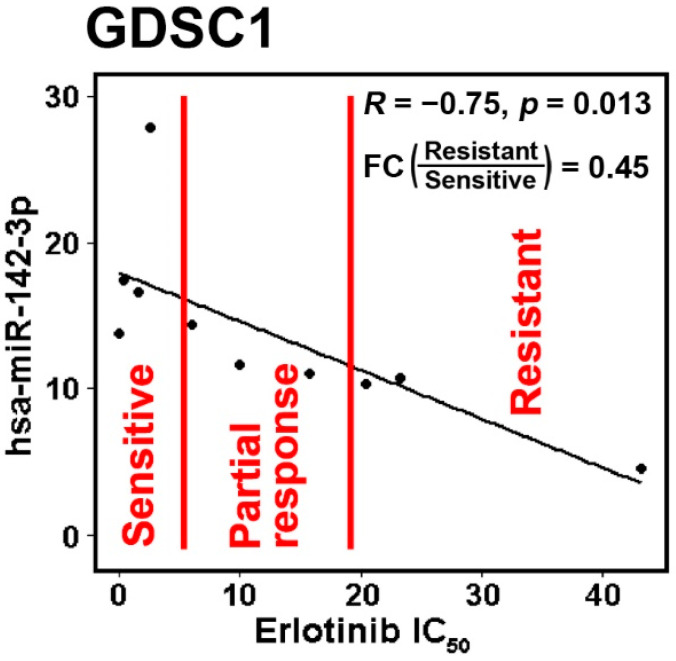
The IC_50_ of erlotinib in the dataset Genomics of Drug Sensitivity in Cancer 1 (GDSC1) had significant negative correlation with hsa-miR-142-3p. The NSCLC cell lines were divided with mean ± 0.5 standard deviation (SD) into three groups: sensitive, partial response, and resistant. The sensitive and resistant groups had significantly differential expression with a fold change (FC) = 0.45.

**Figure 5 ijms-22-07658-f005:**
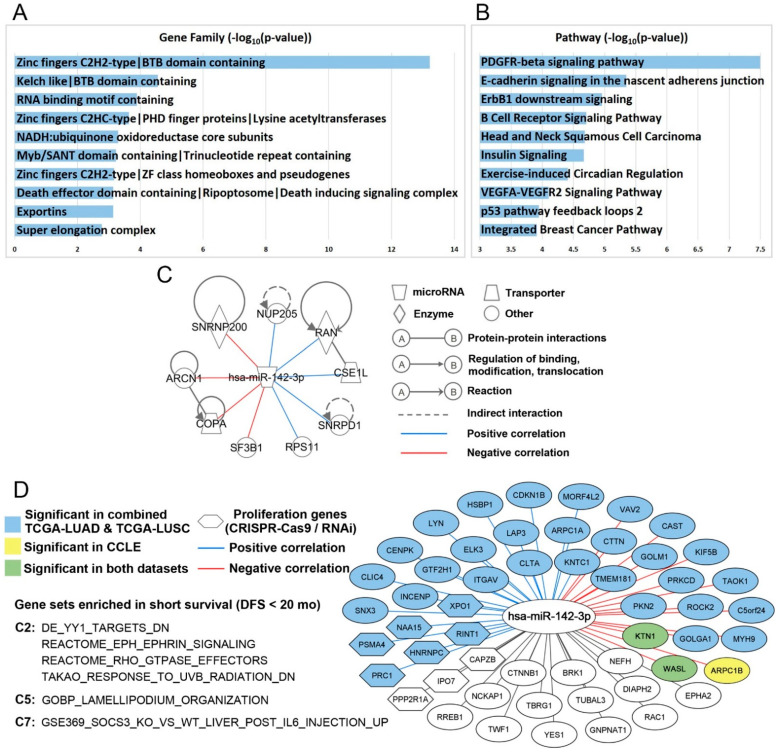
Hsa-miR-142-3p-mediated pathways and functional networks in NSCLC. Top 10 ranked gene families (**A**) and pathways (**B**) of hsa-miR-142-3p target genes in ToppGene functional enrichment analysis. (**C**) Hsa-miR-142-3p target genes with a significant dependency score in 100% of the tested NSCLC cell lines in both CRISPR-Cas9 (*n* = 78) and RNAi (*n* = 92). Detailed information is included in Appendix A, “dependency score” (Appendix A). The experimentally confirmed interactions from IPA were also shown in the plot. The correlation between hsa-miR-142-3p and the target genes was computed in the combined TCGA-LUAD and TCGA-LUSC data. (**D**) Hsa-miR-142-3p target genes that were significantly (*p* < 0.05) enriched in three NSCLC patient cohorts with short-term survival in GSEA in C2, C5, and C7 databases. The colored nodes and edges showed the significant (*p* < 0.05) correlation of hsa-miR-142-3p and the target genes in TCGA and CCLE.

**Table 1 ijms-22-07658-t001:** Summary of demographic and clinical variables for patients in the linked SEER-Medicare database and the clinical cohort used in the population correlation analysis and four validation cohorts from Mary Babb Randolph Cancer Center (MBRCC), Case Western Reserve University (CWRU), TCGA-LUAD, and TCGA-LUSC.

Clinical Variable	Patient Data Used in Population Correlation Analysis		Validation Patient Cohorts			
	SEER-Medicare	Clinical Cohort (Raponi et al.)	MBRCC	CWRU	TCGA-LUAD	TCGA-LUSC
Cancer Stage						
Stage I	12,651 (37.3%)	34 (59.6%)	32 (51.6%)	35 (43.8%)	279 (53.4%)	245 (48.6%)
Stage II	2662 (7.8%)	11 (19.3%)	20 (32.3%)	39 (48.8%)	124 (23.8%)	163 (32.3%)
Stage III	11,514 (34%)	12 (21.1%)	8 (12.9%)	5 (6.3%)	85 (16.3%)	85 (16.9%)
Stage IV	5813 (17.1%)	0 (0.0%)	0 (0.0%)	0 (0.0%)	26 (5.0%)	7 (1.4%)
Unstaged/Other	1257 (3.7%)	0 (0.0%)	2 (3.2%)	1 (1.3%)	8 (1.5%)	4 (0.8%)
Tumor Grade						
Grade 1	1462 (4.3%)	1 (1.8%)	3 (4.8%)	2 (2.5%)	N/A	N/A
Grade 2	13,573 (40.0%)	28 (49.1%)	22 (35.5%)	35 (43.8%)	N/A	N/A
Grade 3	18,202 (53.7%)	28 (49.1%)	27 (43.6%)	32 (40.0%)	N/A	N/A
Grade 4	660 (1.9%)	0 (0.0%)	0 (0.0%)	0 (0.0%)	N/A	N/A
Ungraded/Other	0 (0.0%)	0 (0.0%)	10 (16.1%)	11 (13.8%)	N/A	N/A
Mean Age (σ)	71.1 (7.9)	66.8 (10.7)	67.8 (9.6)	69.5 (9.7)	65.2 (10.0)	67.3 (8.6)
Sex						
Male	22,218 (65.5%)	39 (68.4%)	29 (46.8%)	44 (55%)	242 (%)	373 (74.0%)
Female	11,679 (34.5%)	18 (31.6%)	31 (50.0%)	36 (45%)	280 (%)	131 (26.0%)
Missing	0 (0.0%)	0 (0.0%)	2 (3.2%)	0 (0.0%)	0 (0.0%)	0 (0.0%)

**Table 2 ijms-22-07658-t002:** Total number of significant prognostic miRNAs when considering variable administration of surgery, chemotherapy, and radiation in the SEER-Medicare database and clinical cohort. MiRNAs validated with external patient cohorts from MBRCC, CWRU, TCGA-LUAD, and TCGA-LUSC are in bold.

Treatment	No Chemotherapy	
	Positive Correlation	Negative Correlation
Surgery	miR-520d*, miR-433, **miR-134**, miR-382	miR-328, miR-384, miR-525*
Radiation	miR-433	-
Surgery and radiation	miR-453, miR-520d*, miR-134	miR-197
Any treatment	miR-453, miR-372, **miR-142-3p (miR-142)**, **miR-329 (miR-329-2)**, miR-520d*, miR-433, miR-134, miR-382, miR-493-3p	miR-328, miR-197, miR-384

**Table 3 ijms-22-07658-t003:** Total number of chemosensitive miRNAs when considering administration of specific chemotherapeutic agents in combination with variable administration of surgery and radiation, as evidenced by prolonged disease-specific survival. The miRNAs which produced a significant stratification in the SEER population samples in addition to significance on one or more measures in the NCI-60, MRRCC, CWRU, TCGA, and CCLE are in bold.

Treatment	Cisplatin	Carboplatin	Paclitaxel	Etoposide	Any Chemotherapy
Surgery	**miR-29b (miR-29b-1, miR-29b-2)**	**miR-134**, **miR-142-3p (miR-142)**, **miR-188**, **miR-380-5p (miR-380)**	**miR-134**, **miR-138 (miR-138-2)**, **miR-142-3p (miR-142)**, **miR-188**, **miR-380-5p (miR-380)**	-	**miR-142-3p (miR-142)**, miR-433
Radiation	-	-	miR-154, miR-302a*	**miR-141**, **miR-17-3p (miR-17)**, **miR-17-5p (miR-17)**, **miR-182**, **miR-183**, **miR-19b (miR-19b-1)**, **miR-200c**, **miR-222**, miR-23b	**miR-134**
Surgery And Radiation	-	**miR-134**, **miR-142-3p (miR-142)**	**miR-142-3p**, miR-220	-	**miR-142-3p (miR-142)**
Any treatment	**miR-129 (miR-129-1, miR-129-2)**, **miR-134**, **miR-142-3p (miR-142)**, miR-184, miR-198, **miR-370**, miR-373, miR-379	**miR-134**, **miR-142-3p (miR-142)**, miR-206, **miR-33 (miR-33-a)**, **miR-370**, miR-372	**miR-134**, **miR-142-3p (miR-142)**, **miR-199b**, **miR-370**, **miR-382**	**miR-129 (miR-129-1, miR-129-2)**, **miR-141, miR-142-3p (miR-142)**, miR-184, **miR-218 (miR-218-1)**, miR-220, miR-335, miR-373, **miR-96**	**miR-33 (miR-33-a)**, miR-453, miR-372, **miR-142-3p (miR-142)**, mir-299-3p, **miR-329 (miR-329-2)**, miR-520d*, miR-519a, miR-494, miR-433, **miR-134**, **miR-485-5p (miR-485)**, miR-518c*, miR-493-3p

**Table 4 ijms-22-07658-t004:** Total number of chemoresistant miRNAs when considering administration of specific chemotherapeutic agents in combination with variable administration of surgery and radiation, as evidenced by shortened disease-specific survival. The miRNAs which produced a significant stratification in the SEER population samples in addition to significance on one or more measures in the NCI-60, MRRCC, CWRU, TCGA, and CCLE are in bold.

Treatment	Cisplatin	Carboplatin	Paclitaxel	Etoposide	Any Chemotherapy
Surgery	**miR-126***, miR-136, miR-181b, **miR-181c**, **miR-196a (miR-196a-2)**, **miR-331**, **miR-375**, miR-424, **miR-92 (miR-92a-1, miR-92b)**	**miR-126**, **miR-192**, miR-195, miR-384	miR-189, miR-192, miR-197, miR-301, miR-328, **miR-331**, miR-384	-	miR-197, miR-384
Radiation	miR-384	-	miR-189	miR-150, miR-155, miR-192, miR-337, miR-98	miR-328
Surgery and Radiation	-	**miR-192**, miR-384	miR-192, miR-197, miR-328, **miR-331**, miR-384	miR-223	miR-328, miR-197, miR-384
Any treatment	**miR-132**, miR-181b, miR-30c, miR-30e-3p, miR-324-3p, **miR-331**, miR-339, miR-384	**miR-126**, miR-195, miR-224, miR-384	**miR-126**, miR-224, miR-328, **miR-331**, miR-361, miR-384	**miR-132**, miR-133b, miR-155, miR-197, miR-208, miR-214, miR-324-3p, miR-374, miR-423	miR-328, miR-361, miR-511, miR-197, **miR-125a**, miR-384, **miR-126**

**Table 5 ijms-22-07658-t005:** Hsa-142-3p targeted proliferation genes (shown in Figure 5C) with a significant differential expression (*p* < 0.05; two-sample *t*-tests) in sensitive versus resistant CCLE NSCLC cell lines to specific drugs. Drug activity measurements include IC_50_, EC_50_, ln(IC_50_), and ln(EC_50_). Red font indicates that the gene has a higher expression in the resistant cell lines than in the sensitive cell lines. Blue font indicates the gene has a lower expression in the resistant cell lines than in the sensitive cell lines.

Drugs	PRISM	GDSC1	GDSC2
Cisplatin	*SNRNP200*	*SF3B1* , *RAN*	*CSE1L*
Erlotinib	*SNRNP200*	*CSE1L*	
Gefitinib		*COPA*	
Paclitaxel			*NUP205*
Pemetrexed		*RAN* , *SNRPD1*	

## Data Availability

Access to data available from public domains is provided in the manuscript. The NCI SEER-Medicare dataset requires a license permit for data access and usage.

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
