# Peer review of "Identification of Prognostic and Chemopredictive microRNAs for Non-Small-Cell Lung Cancer by Integrating SEER-Medicare Data"

_ijms, 2021, doi:10.3390/ijms22147658_

Round 1

Reviewer 1 Report

The study is very interesting and speculative, but are there some applications of these results  that could be applied in routine practice ?

Since previous chemotherapy markers (e.g. RRM1, ERCC1, TS) failed to enter in practical use, have the authors tested some of the Hsa-miRNA-142-3p target genes, namely NUP205, RAN, CSE1L, SNRPD1, RPS11, SF3B1, COPA, ARCN1 and/or SNRNP200, in tissue from FFPE or fresh tumors, at least by IHC ?

Author Response

Reviewer 1

Comments and Suggestions for Authors

Reviewer: The study is very interesting and speculative, but are there some applications of these results that could be applied in routine practice?

Authors: We thank the reviewer for the positive comment. This study presents a novel methodology to correlate a clinical cohort (Raponi et al) with SEER-Medicare data to identify miRNAs that had concordant association with patient survival in both the clinical cohort and the population correlation. Based on the patient treatment information documented in the SEER-Medicare data, prognostic miRNAs were identified in patients who did not receive any chemotherapy, and chemo-predictive miRNAs were identified in patients who received chemotherapy in the population correlation. These identified miRNAs were further validated with patient samples collected from several US hospitals (MBRCC/CHTN, and CWRU), as well as public data including TCGA, NCI-60, and CCLE. The miRNAs validated with the external patient cohorts and public data have the potential to be used in clinical practice for prognosis and the selection of chemotherapy in NSCLC treatment. Particularly, hsa-miR-142-3p was identified as a good prognosis and chemosensitivity biomarker that was validated in every dataset in this study, with promising results for clinical utility upon further prospective evaluation. The identified hsa-miR-142-3p mediated regulatory pathways and networks in NSCLC reveal important mechanistic insights with potential therapeutic targets for invention.

Reviewer: Since previous chemotherapy markers (e.g. RRM1, ERCC1, TS) failed to enter in practical use, have the authors tested some of the Hsa-miRNA-142-3p target genes, namely NUP205, RAN, CSE1L, SNRPD1, RPS11, SF3B1, COPA, ARCN1 and/or SNRNP200, in tissue from FFPE or fresh tumors, at least by IHC?

Authors: We thank the reviewer for this comment. We now add the association of these genes with chemo-response in CCLE NSCLC cell lines in Table 5. In our future study, we will examine these genes in NSCLC patient frozen and FFPE tissues using qRT-PCR, ELISA, and/or IHC to further substantiate their clinical relevance in treatment response.

Reviewer 2 Report

In the manuscript entitled "Identification of prognostic and chemo-predictive microRNAs for non-small cell lung cancer by integrating SEER-Medicare data" by Ye et al., the authors describe the identification of hsa-miR-142-3p expression as prognostic marker and predictive of chemosensitivity in NSCLC patients. Several publically available datasets were re-analysed. Further in silico prediction of hsa-miRNA-142-3p targets was performed and network analyses of these target genes were done.

Comments:

For the reader it is hard to understand how the authors describe a prognostic miRNA in a dataset where miRNA expression was not analysed (SEER). Why do you need SEER data without miRNA expression information when you have 4 datasets with miRNA expression information? Also the labeling in Figure 2 C and D ("Over-expression" and "Under-expression") is misleading because the authors do not have information about miR-142-3p expression in this dataset. This needs to be re-written to increase clarity.

Is "Chemotherapy" in CWRU and MBRCC datasets comparable to information from SEER data shown in Figure 2C?

The authors describe the MBRCC and CWRU datasets as NSCLC datasets. The initial miRNA identification was done using the Raponi cohort which consists of squamous cell carcinoma patients only. Because lung adenocarcinomas and lung squamous cell carcinomas are biologically different, the validation datasets should also be split based on the histological subtypes.

Figure 2: The title of this figure is "Over-expression of hsa-miR-142-3p correlates with prolonged survival and chemo-sensitivity in NSCLC patients". This is misleading because based on E/F and G/H the survival difference is independent from hsa-miR-142-3p expression.

Where the p-values corrected for multiple testing?

miRNA target genes were identified using TarBase. This procedure should be done using multiple algorithms to increase sensitivity.

Author Response

Comments and Suggestions for Authors

In the manuscript entitled "Identification of prognostic and chemo-predictive microRNAs for non-small cell lung cancer by integrating SEER-Medicare data" by Ye et al., the authors describe the identification of hsa-miR-142-3p expression as prognostic marker and predictive of chemosensitivity in NSCLC patients. Several publically available datasets were re-analysed. Further in silico prediction of hsa-miRNA-142-3p targets was performed and network analyses of these target genes were done.

Comments:

Reviewer: For the reader it is hard to understand how the authors describe a prognostic miRNA in a dataset where miRNA expression was not analysed (SEER). Why do you need SEER data without miRNA expression information when you have 4 datasets with miRNA expression information? Also the labeling in Figure 2 C and D ("Over-expression" and "Under-expression") is misleading because the authors do not have information about miR-142-3p expression in this dataset. This needs to be re-written to increase clarity.

Authors: This study sought to identify prognostic and chemo-predictive miRNAs to improve treatment for NSCLC. The patient cohorts we collected from US hospitals (MBRCC/CHTN and CWRU) had limited sample size. The small sample size made it infeasible to correlate miRNA expression to the clinical outcome of specific chemotherapy (i.e. cisplatin, carboplatin, paclitaxel, alimta/pemetrexed, etc) in a particular NSCLC subtype (i.e., adenocarcinoma or squamous cell carcinoma). The public TCGA data did not provide chemotherapy information. The drug activity data in the NCI-60 and CCLE panels are in vitro and do not always represent the results in patients.  To overcome these limitations, this study developed a novel methodology to correlate a clinical cohort (Raponi et al) with SEER-Medicare data to identify miRNAs that had concordant association with patient survival in both the clinical cohort and the population correlation. Based on the patient treatment information documented in the SEER-Medicare data, prognostic miRNAs were identified in patients who did not receive any chemotherapy, and chemo-predictive miRNAs were identified in patients who received chemotherapy in the population correlation. These identified miRNAs were further validated with patient samples collected from several US hospitals (MBRCC/CHTN, and CWRU), as well as public data including TCGA, NCI-60, and CCLE.

Figure 2C and D were corrected. We thank the reviewer for pointing it out.

Reviewer: Is "Chemotherapy" in CWRU and MBRCC datasets comparable to information from SEER data shown in Figure 2C?

Authors: “Chemotherapy” in CWRU and MBRCC refers to any chemotherapy, including cisplatin, carboplatin, paclitaxel, pemetrexed, etc. In Figure 2C, only patients who received paclitaxel in SEER-Medicare were included in the analysis. This clarification is now added in the manuscript.

Reviewer: The authors describe the MBRCC and CWRU datasets as NSCLC datasets. The initial miRNA identification was done using the Raponi cohort which consists of squamous cell carcinoma patients only. Because lung adenocarcinomas and lung squamous cell carcinomas are biologically different, the validation datasets should also be split based on the histological subtypes.

Authors: The patient cohorts we collected from US hospitals (MBRCC/CHTN and CWRU) had limited sample size. The small sample size made it infeasible to correlate miRNA expression to the clinical outcome of specific chemotherapy (i.e. cisplatin, carboplatin, paclitaxel, alimta/pemetrexed, etc) in a particular NSCLC subtype (i.e., adenocarcinoma or squamous cell carcinoma). The TCGA data was analyzed according to NSCLC subtypes, TCGA-LUSC and TCGA-LUAD, respectively.

Reviewer: Figure 2: The title of this figure is "Over-expression of hsa-miR-142-3p correlates with prolonged survival and chemo-sensitivity in NSCLC patients". This is misleading because based on E/F and G/H the survival difference is independent from hsa-miR-142-3p expression.

Authors: In Figure 2E, F, G, H, the NSCLC patients in MBRCC and CWRU cohorts were stratified based on hsa-miR-142-3p expression. Patients with a higher expression of hsa-miR-142-3p had longer survival than those with a lower expression of hsa-miR-142-3p in specific clinical settings, i.e., receiving chemotherapy (E, G) or without chemotherapy (F, H). In Figure 2B, expression of hsa-miR-142-3p was positively correlated with chemosensitivity to paclitaxel in the NCI-60 cell lines. Together, the results showed that over-expression of hsa-miR-142-3p correlates with prolonged survival and chemo-sensitivity in NSCLC patients.

Reviewer: Where the p-values corrected for multiple testing?

Authors: The ToppFun tool used the false discovery rate (FDR) multiple correction method with enrichment significance cutoff level of 0.05 in the gene list enrichment analysis. In Gene Set Enrichment Analysis (GSEA), the FDR q-value was used to select candidate gene sets. An FDR q-value < 0.05 indicates statistical significance. It is now added in the manuscript, section 4.11.

Reviewer: miRNA target genes were identified using TarBase. This procedure should be done using multiple algorithms to increase sensitivity.

Authors: TarBase includes experimentally confirmed target genes for a miRNA. Other algorithms such as TargetScan, miRbase, miRDB, PicTar, and miRNA target gene prediction from abcam provide predicted target genes. This study sought to investigate experimentally confirmed target genes of hsa-miR-142-3p, not computationally predicted target genes. Therefore, TarBase v7.0 and v8.0 were used to retrieve experimentally confirmed target genes of hsa-miR-142-3p. Refences of the original studies proving an interaction between the reported target gene and hsa-miR-142-3p were provided in the manuscript.

Reviewer 3 Report

Dear Authors,

The Manuscript by Qing Ye et al, entitled “Identification of prognostic and chemo-predictive microRNAs for non-small cell lung cancer by integrating SEER-Medicare data” provides an interesting, well written approach integrating SEER-Medicare data combined with comprehensive external validation which can identify miRNAs with consistent expression patterns   in   tumor   progression,   with   potential   implications   on   prognosis   and   prediction   of    chemoresponse in large NSCLC patient populations. Therefore, I strongly recommend the publication after small, minor revisions.

  1. Since “chemo-predictive” is used in the title, correct “chemopredictive”:

page 22 line 722,

page 23 line 728

page 24 line 735

page 25 line 744.

  1. Line 181: Correct “tubulin”
  2. Line 185: Explain the “IPA” abbreviation, which is not mentioned until page 20.
  3. Line 427: Correct miR-134
  4. Line 429: Lowercase “In”

Author Response

We thank the reviewer for the positive comments. We have made all corrections recommended by the reviewer.

Round 2

Reviewer 2 Report

It is still very unclear how the authors can propose the prognostic potential of a miRNA in a patient population without miRNA expression data. The authors did not re-write the corresponding paragraphs to increase clarity. The authors mention that "Specifically, the AJCC TNM classifications and tumor grade were used to partition patients into disjoint groups indicative of tumor progression, with the average survival being calculated for each group". It is obvious that the clinical stage of NSCLC patients is associated with patient survival. So why do you need a miRNA which expression status is completely unknown in the SEER patient cohort as additional factor?

ad adjustment of p-values: It is clear that bioinformatic tools (ToppFun, GSEA) properly adjust p-values for multiple testing. However, the many p-values resulting from Cox regression analyses in this study were obviously not adjusted for multiple testing, thus, the number of false positive miRNAs is likely high. This is an issue and should be discussed properly.

Author Response

There has been reported success in identifying chemo-predictive genes by correlating mRNA expression profiles in solid tumors of the SAGE database with patient survival in the SEER data [26]. In the study by Stein et al [26], genes differentially expressed between solid tumors and cell lines were first selected from the SAGE database without being adjusted for multiple testing. To control false discovery, gene expression in solid tumors of the SAGE database was correlated with a 5-year survival of patients with a distant disease in the SEER data by different tumor types. Stein et al [26] used the SEER 5-year survival data as a surrogate for chemosensitivity, acknowledging that factors other than chemosensitivity also influence patient survival. Here, since we focused on NSCLC, the approach presented by Stein et al [26] was adapted in the following way. A composite tumor progression indicator based on AJCC TNM cancer stage and tumor grade was used as a surrogate to correlate miRNA expression with SEER-Medicare patient survival, recognizing that the cancer stage is a strong indicator of NSCLC survival. Specifically, the AJCC TNM cancer staging classifications and tumor grade (G) were used to partition SEER-Medicare patients into disjoint groups (i.e. T=1; N=0; M=0; G=1) indicative of tumor progression, with the average survival being calculated for each group. Patients in the training clinical cohort from Raponi et al [9] were partitioned in a similar manner. Linear regression was be used to estimate the association between average miRNA expression (per group) in the training clinical cohort from Raponi et al [9] and corresponding average survival in SEER-Medicare patients. The results were compared with those from independent clinical validation cohorts in the public domain, including our cohorts upon publication. Drug activities of commonly used chemotherapy, together with miRNA and mRNA profiles in the NCI-60 and CCLE cell lines, were used to corroborate the results on chemo-response prediction. Similar to Stein et al [26], prognostic and chemo-predictive miRNAs were first selected using univariate Cox model and Kaplan-Meier analysis in training clinical cohort from Raponi et al [9] without multiple testing. The false discovery was controlled by 1) population correlation with SEER-Medicare data with linear regression and corroboration with Kaplan-Meier analysis, and 2) with additional external validation using our collected patient cohorts, as well as TCGA, NCI-60, and CCLE data.

The above description is now added to the Discussion.